# Beyond Maslow's Pyramid: Introducing a Typology of Thirteen Fundamental Needs for Human-Centered Design

**Pieter Desmet \* and Steven Fokkinga**

Faculty of Industrial Design Engineering, Delft University, 2628 CE Delft, The Netherlands;
s.f.fokkinga@tudelft.nl
\* Correspondence: p.m.a.desmet@tudelft.nl

**Abstract:** This paper introduces a design-focused typology of psychological human needs that includes 13 fundamental needs and 52 sub-needs (four for each fundamental need). The typology was developed to provide a practical understanding of psychological needs as a resource for user-centered design practice and research with a focus on user experience and well-being. The first part of the manuscript briefly reviews Abraham Maslow's pioneering work on human needs, and the underlying propositions, main contributions and limitations of his motivational theory. The review results in a set of requirements for a design-focused typology of psychological needs. The second part reports on the development of the new typology. The thirteen needs were selected from six existing typologies with the use of five criteria that distinguish fundamental from non-fundamental needs. The resulting typology builds on the strengths of Maslow's need hierarchy but rejects the hierarchical structure and adds granularity to the need categories. The third part of the paper describes three examples of how the need typology can inform design practice, illustrated with student design cases. It also presents three means for communicating the need typology. The general discussion section reflects on implications and limitations and proposes ideas for future research.

**Keywords:** psychological needs; user-centered design; design for experience; design for well-being

## 1. Introduction

*Much recent design has satisfied only evanescent wants and desires, while the genuine needs of man have often been neglected. The economic, psychological, spiritual, social, technological, and intellectual needs of a human being are usually more difficult and less profitable to satisfy than the carefully engineered and manipulated "wants" inculcated by fad and fashion. Victor Papanek, 1971* [1] (p. 15).

If one thing can be said about design, it is that its relevance to the individual and to humanity at large—its purpose, meaning, success or failure—depends on the extent to which it satisfies people's needs. Design is often framed as a problem-solving discipline. The root of these problems is always some kind of obstacle to need fulfilment, be it the need for safety, friendship, autonomy, purpose, or any other need. Products, systems, buildings, technology, and all other 'artificial' phenomena envisioned and built by people are instruments for need satisfaction. Smartphones help us connect to the people who are important to us (need for relatedness), cars take us where we want to go (need for autonomy), and ready-made meals give us nutrition with very little effort (need for comfort). Needs are the fundament of our motivation system, and all human activity is essentially fuelled by the aspiration of need fulfilment [2]. This includes the activity of designing: all design practices are articulations of human needs. This means that designs do more than merely serve the needs of end users—they

affect the need fulfilment of every single person (including the designers) involved in their ideation, development, execution, distribution, and usage practices.

What do we mean by human needs? In this paper, we adopt the definition given by Deci and Ryan, who proposed that needs are the basic requirements for the functioning of an organism. "Human needs specify innate psychological nutriments that are essential for ongoing psychological growth, integrity, and well-being" [3] (p. 229). From their perspective, needs play a necessary part in human well-being and advancement. Individuals can only fully develop if all their basic needs are satisfied (at least to some degree), whether or not they are consciously valued, and none can be neglected without significant negative consequences [4]. Beyond their crucial role of as nutriments for well-being, needs are also a strong direct source of meaning and pleasure (and displeasure): events and activities that fulfil our needs are experienced as meaningful and pleasurable [5]. If you take a moment to reflect on the activities that you find personally most enjoyable and meaningful, you will probably find that these activities fulfil at least one of your basic needs — most probably multiple needs. Why do we enjoy having dinner with friends? Because it fulfils our needs for friendship and stimulation. Why do we value traveling? Because it fulfils our needs for personal growth and autonomy.

The importance of human needs is generally recognized in design research and practice. Hassenzahl and colleagues found a positive relationship between need fulfilment and enjoyable experiences with technological artifacts [6], see also [7,8]. Informed by these findings, they proposed that human needs provide a solid fundament for positive experiences created and mediated through design and technology. Likewise, in our own research, we found a direct relationship between the emotions a person experiences when seeing, using, or owning consumer products, and the degree to which these products enable that person to satisfy his or her needs (e.g., [9,10]). In design practice, need profiles can support a systematic approach to design for positive experiences and subjective well-being [6,11,12].

An effective start to understanding and manipulating a phenomenon is to break it down into meaningful categories or dimensions, which often take the form of a typology [13,14]. They can be used as analytic tools, to form and refine concepts, draw out underlying dimensions, and create categories for measurement [15]. In design research and practice, need typologies can provide a shared language about user needs, they can help making need profiles, and they can serve as a lens to evaluate or measure the needs fulfilled (or thwarted) by existing designs or new concepts. What typology of human needs is best suited for design research and practice? This is the central question addressed in this paper. A variety of typologies are available in the literature. Some are concise, discerning needs most vital to human functioning. A good example is the well-known typology of Deci & Ryan, which includes three fundamental needs: autonomy, relatedness and competence [16]. Similarly concise typologies list five needs [17] (based on [5]), ten [5,18], fifteen [19], eighteen [20], or twenty-three [21]. Other typologies are more extensive and aim to provide full overviews. The most extensive typology, to our knowledge, is the overview of 161 needs completed by Talevich and colleagues [22]. Other examples include 56 [23], or 135 [24]. Even though there is a substantial overlap of needs included in these typologies, there are also some notable differences. None of them can be considered the 'best' because the value of a typology depends on its intended application, which differs between academic fields.

The need typology that is most often referred to in the field of design is probably psychologist Abraham Maslow's famed hierarchy of needs. Maslow [25] distinguished between five basic need categories (physiological, safety, social, esteem, and self-actualization). Over the years, we have observed that many of our design students favour this need overview, applying it wholeheartedly in their user-centered design projects. Its widespread success has been ascribed to its simplicity, the ease with which it can be taught or explained, its intuitive pitch, and the fact that it looks great in a PowerPoint (see e.g., [26,27]). The concise and organized overview provides a direct and actionable understanding of the complexities of human motivation. At the same time, there is a growing awareness that Maslow's theory does not meet the requirements for scientific rigor, that it is outdated, and perhaps even misleading. The truth seems to lie somewhere in the middle. While, indeed, an impressive amount of empirical data has shown that some of the theory's underlying propositions are incorrect,

some of the other propositions have been convincingly replicated and supported. We propose that when Maslow's hierarchy of needs is used, it should be done with an understanding of the validity of its theoretical underpinnings. In our experience, however, most design students (and many researchers) refer to the hierarchy without specifying which of the underlying propositions they accept or adhere to. Moreover, the usefulness of any typology of needs depends on the purpose of its application [13]. Maslow's need theory should be applied and referred to with the full awareness that it was not developed to inform and inspire design and design research.

This paper aims to contribute to an accurate understanding of human needs as an inspiring foundation for user-centered design practice and research. We first briefly discuss Maslow's Hierarchy of Needs, its intentions and underlying propositions, its main contributions, and its limitations. Next, we describe the requirements for a design-focused revision of the hierarchy (to overcome its limitations), and we introduce a typology of thirteen fundamental needs based on those requirements. Our typology builds on the strengths of Maslow's need hierarchy but rejects its hierarchical structure and adds granularity to each need category. In the second part of the paper, we describe some application possibilities illustrated with three student design cases, and three means for disseminating the need typology. In the general discussion, we reflect on implications and limitations, and propose ideas for future research.

## 2. Maslow's Hierarchy of Needs

Abraham Maslow (1908–1970) is recognized as one of the most eminent psychologists of the 20th century [28]. His work ranks #14 in the list of most frequently cited in introductory psychology textbooks [29]. He was one of the founders of the humanistic approach to psychology, whose aim is to study the positive potential of human beings. The humanistic perspective starts from the idea that all people have a natural drive for personal growth, and that the ultimate goal of living is to realize one's full potential—to be all one can fully be. In the early twentieth century, this voice was novel to the scientific discourse. At the time, psychology was dominated by two paradigms: behaviourism, which aimed to reduce human functioning to simple input-output mechanisms; and psychoanalysis, which predominantly focused on abnormal psychological processes and problematic behaviours. In contrast to these approaches, Maslow was more interested in understanding positive behaviour and what it is that people do that makes them happy. He dedicated decades of study to this positive behaviour, and his scientific legacy has been the foundation for the current positive psychology movement [30].

The origin of Maslow's theory was a simple question: 'What motivates humans?' His theory proposes that all human activity is (directly or indirectly) motivated by innate needs, which can be physiological (such as the need for water and oxygen) or psychological (such as the need for love and independence). The motivational theory crystallized as the Hierarchy of Needs [25,31,32], one of the most influential motivational theories in the history of psychology [33]. Conceived almost 80 years ago, it is still widely used and passed on, mostly in applied field such as design, education, healthcare, management, communication, and marketing. In the body of his work, Maslow puts forth a number of theses about needs, motivation, and behaviour, which we summarize here as eight key propositions:

1)  Human behaviour is **motivated**. Behaviour cannot be explained as mere reactions to external events and conditions.
2)  A person's motivation is driven by their **needs**. Individuals strive to fulfil their needs and to avoid need dissatisfaction.
3)  A broad diversity of behaviour can be explained with a relatively **small set** of fundamental needs.
4)  Fundamental needs are **inborn** and universal; they apply to all humans across ages and cultures.
5)  For a human to survive, function, and flourish, **all needs** should be fulfilled, at least to some degree.
6)  Only needs that are **not yet satisfied** motivate behaviour; when a need is satisfied, it ceases to be a motivator.
7)  Needs are organized in a **hierarchy**, addressed in an order, from basic to complex.

8) When a lower-order need is gratified, this will prompt the **activation** of a next-level need. Next-level needs remain dormant until lower-order needs are satisfied.

The heart of Maslow's motivational theory is his hierarchy of needs, often visualized as a pyramid (Figure 1). The hierarchy includes five categories, starting with physiological needs at the base, moving upward to safety needs, social needs, esteem (or ego) needs, and finally, self-actualization needs. In brief, Maslow proposed that humans have to satisfy their biological needs first, then they can seek order and predictability within their lives, a sense of personal worthiness and importance, love and affection with important others, and finally, a sense that they are moving toward an ideal version of themselves. Table 1 gives a more elaborate overview of the five need categories.

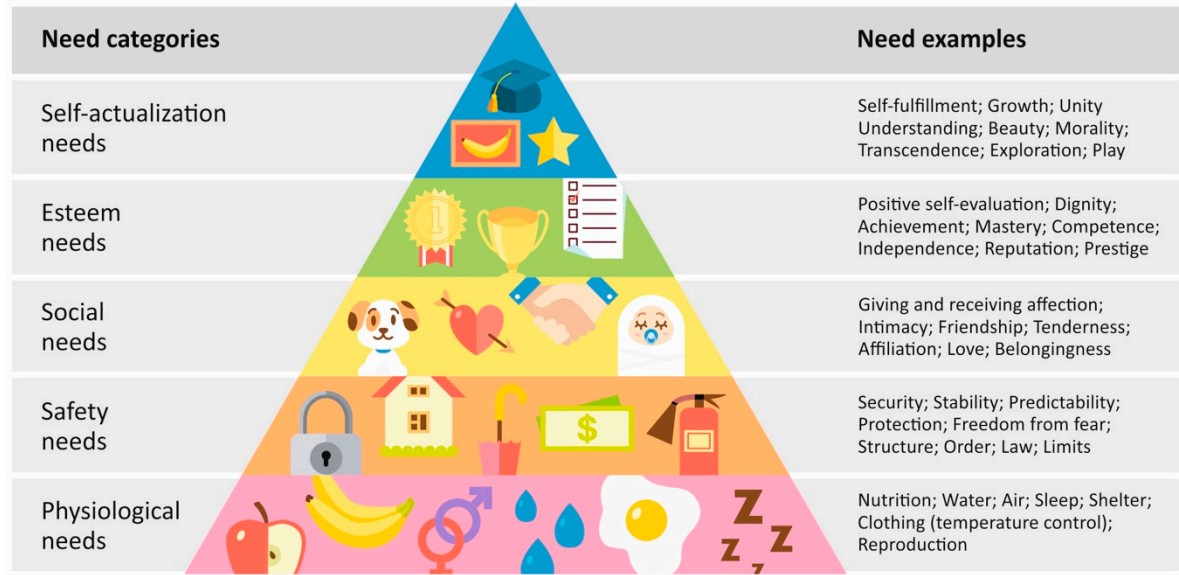

**Figure 1.** Pyramid representation of Maslow's Hierarchy of Needs (image based on [31]). Even though the popularity of Maslow's theory is partly attributed to the iconic appeal of the pyramid (or, more accurately, the triangle), Maslow himself never used the triangle image. In a recent paper, Bridgman, Cummings, and Ballard [34] traced the origins of the triangle visualisation to a manuscript of the consulting psychologist Charles McDermid [35]. After that, the image was popularized by management textbooks.

**Table 1.** The five need categories in Maslow's Hierarchy of Needs.

| Need Category | Description |
|---|---|
| 5. Self-actualization Needs | Once all previous needs have been met, an individual can direct his or her focus toward the 'development of the self'. This category includes the highest-level needs that one could satisfy, such as the need to maximize one's potential, or the needs for personal growth, creativity, morality, and meaning making. |
| 4. Esteem Needs | Esteem needs fall into two categories. The first contains needs for validation from others, such as the need for status, respect, recognition, and reputation. The second includes needs for positive self-evaluation, such as the need for competence, confidence in ability, accomplishment, and skills mastery. |
| 3. Social Needs | Social needs include the need for belonging, love, intimacy, and affection. Relationships with friends, romantic partners, and families fulfil this need, as does involvement in communities and social or religious groups. |
| 2. Security Needs | Security needs are psychological, such as the need for a safe family environment, steady employment, a safe neighbourhood, and a stable financial situation. |
| 1. Physiological Needs | Physiological needs are the basic needs of any living organism: the requirements for the body to survive, such as the need for water, oxygen, food, and sleep. In general, physiological needs influence behaviour through direct desires or cravings. |

Maslow termed the four lower categories as 'deficit needs', by which he meant that a person will experience negative physiological and psychological consequences if they are not fulfilled. They are activated by deprivation, which means that they will remain dormant as long as they are fulfilled. For example, our social needs will not motivate our actions as long as we are surrounded by loved ones. But moving to a new country will probably awaken social needs, which, in turn, motivate us to initiate new relationships. In contrast, the highest category, needs related to self-actualization, is not activated by deprivation. Maslow called them 'growth needs' (or being needs) that do not stem from a lack of something but rather from an inborn desire to grow as a person. They will continue to be felt when they are satisfied and may even become stronger once engaged.

While the need categories are essentially universal (and shared by people of all ages and all backgrounds), how they translate into everyday life varies greatly from person to person, depending on age, personality, context, and culture. Take, for example, the need for safety. For a three-year old, it may manifest as the need for a gentle caregiver, for a teenager it may manifest as the need for an accepting peer group, for an adult as the need for job stability, and so on [36]. Maslow pointed out that the self-actualization category especially is expressed as highly individual actions. In his words, 'In one individual it may take the form of the desire to be an ideal mother, in another it may be expressed athletically, and in still another it may be expressed in painting pictures or in inventions' [25] (pp. 382–383). Moreover, there is not a one-to-one relationship between human needs and behaviour, because most behaviour is motivated by multiple needs simultaneously. Having dinner with a friend, for example, can simultaneously satisfy physiological needs, social needs, and self-actualization needs.

## 3. Strengths and Limitations of the Hierarchy of Needs

Maslow's motivational theory has been criticized for its lack of empirical support and the inadequate operationalization of its concepts ([37] for a discussion, see [38]). In addition, many scholars have questioned the practicality and reality of the hierarchy, the process by which people proceed through the hierarchy, and its relevance and applicability to modern society [38]. At the same time, the theory has introduced some crucial ideas that have withstood the test of time and are highly useful for design research and practice. The scope of this paper does not allow us to revisit every single critique, so here we will discuss two of the hierarchy's best-established contributions and two of its oft-discussed limitations. We consider these four to be quite important when applying the Hierarchy of Needs in design research and practice.

### 3.1. Contribution 1: Human Needs are Universal

Tay and Diener have rigorously tested Maslow's theory by analysing more than sixty thousand individuals from 123 countries, representing every major region of the world [4]. The results of their study conclusively support the view that human behaviour is motivated by needs (propositions 1 & 2), and that fundamental needs are universal and exist regardless of cultural differences (propositions 3 & 4). Likewise, Ryff and Keyes [39] and Ryan and Deci [40] support the idea that there are universal human needs 'wired into humans' and that fulfilment of them is likely to enhance a person's feelings of well-being.

### 3.2. Contribution 2: Fulfilling Fundamental Needs Contributes to Well-Being

The study by Tay and Diener also found a strong association between need fulfilment and subjective well-being across world regions [4]. This supports the idea that well-being requires a basic fulfilment of all fundamental needs (proposition 5). Need satisfaction has been shown to predict well-being outcomes both in general, with people who report greater overall need satisfaction also reporting greater well-being, and on a day to day basis, with daily fluctuations in need fulfilment predicting daily fluctuations in well-being (for a review, see [41]). Each need will enhance well-being to some extent when it is fulfilled, and lack of fulfilment of one need cannot be compensated by 'over-fulfilment' of another. In the words of Tay and Diener: 'Like vitamins, each of the needs is individually required, just

as having much of one vitamin does not negate the need for other vitamins' [4] (p. 355). All needs independently contribute to a person's well-being. Just because one has, for example, a large amount of food and safety, it does not follow that one's need for social support diminishes.

### 3.3. Limitation 1: Hierarchal Order

The idea that needs are organized in a hierarchy (proposition 7) has been disproven. People can be motivated by higher-order needs independent of the degree to which their lower-order needs are satisfied. For example, studies have shown that people who live in poverty are still capable of pursuing higher-order needs such as love, esteem, and beauty, even though they are struggling to achieve basic physiological needs such as food and shelter. Therefore, it is incorrect to assume that higher-order needs remain dormant until lower-order needs have been satisfied [37,42,43]. You can enjoy time with your friends even when you are hungry. Maslow himself acknowledged this in his later writings. While he [25] initially stated that individuals must satisfy lower level deficit needs before progressing toward meeting higher level growth needs, he later clarified that satisfaction of a need is not an 'all-or-none' phenomenon: "In actual fact," Maslow states, "most members of our society who are normal and partially satisfied in all their basic needs and partially dissatisfied in all their basic needs at the same time" [31] (p. 69–70). A second problem with the hierarchal order is that it is biased by ethnocentricity. The idea that the highest-level human need is self-actualization reflects the ideology of individualistic societies. Maslow's theory emerged from an American cultural setting, in which the individual is the point of reference and the realization of the 'individual' is the highest goal. Several authors have stressed that this Western ideology does not apply to other societies, such as those of Africa and Asia, in which collectivism and conviviality are central, and where social needs out-prioritize self-focused needs [44,45]. Moreover, social scientists have convincingly shown that any individual's wellbeing (also those in individualistic societies) is influenced by social factors, such as social cohesion and acceptance (i.e., being integrated into one's community), and social actualization (i.e., being able to contribute to society and the social good) [46,47].

### 3.4. Key Limitation 2: The Problem with Self-Actualization

While it is a key ingredient of the model, the concept of self-actualization is problematic. To start, it was based on samples of what Maslow considered to be self-actualized individuals, a cohort predominantly limited to highly educated white men (such as Abraham Lincoln and Albert Einstein). Although in a later stage, Maslow [32] also studied self-actualized women (such as Eleanor Roosevelt and Mother Teresa), they comprised a small proportion of his sample. This makes it difficult to generalize his theory to individuals of other social classes, ages, genders, and education levels. In addition, the category of self-actualization is very ambiguous. Like all layers in the hierarchy, self-actualization is not a need as such but a category of needs. The category is too broad to be conceptually clear; Maslow seems to have used it as a container for every other human need that does not fall into the other four categories. Some needs he mentioned are the need for self-development and learning, spirituality and transcendence, wisdom, creativity, beauty, justice, play, virtue, autonomy, individuality, and many more. These needs seem so diverse that putting in them one category does not do them justice, especially from a design perspective. Moreover, even though it is attractive at first sight, the concept of self-actualization has been shown to be too vague in and of itself to function as a guiding factor in understanding human motivation [48].

In sum, we can assert that Maslow's propositions 1 to 5 have withstood the test of time (and empirical validation), while propositions 6 to 8 have proven faulty. This has two implications for design research and practice. The first is that we can assume there is a limited set of innate fundamental needs, and that these needs apply to all humans. We can also assume that all of these needs contribute to the well-being of an individual. At the same time, we have to reject the idea that there is a particular order of relevance in these needs, or that the relevance of one need depends on the degree of fulfilment of another. This means that an accurate need typology provides a clear overview of needs, while it

avoids making speculations about which need is relevant for which individual in which situation. The second implication is that we should realize that self-actualization is not a need itself but a state reached when self-actualization needs are fulfilled—while it is also not yet clear what those needs are. This means that a need typology should clarify what fundamental needs are represented by the concept of self-actualization and extend it from a mere individualistic concept to one that also includes a social focus on well-being. While this is particularly relevant for self-actualization, it basically applies to all layers: they are need categories, and it is not yet clear which fundamental needs they embody.

## 4. Developing an Improved Typology of Fundamental Needs

Our objective was to develop a typology that builds on the strengths of the hierarchy of needs, circumvents its limitations, and operationalizes need categories as actual, fundamental needs. Because this is a process of typology development, first we will present the fundamental components of any typology. These fundamentals will act as the basis for our later explanations of the key considerations we took into account while developing our need typology.

First of all, a definition: a typology describes a particular concept, which is the phenomenon of interest. It reduces complexity by categorizing objects (which are qualities or instances of the concept) into types. The objects, which can be tangible (e.g., home appliances) or intangible (e.g., personality traits), are grouped together into a type on the basis of shared characteristics. As a form of theory, typologies are descriptive: they describe a phenomenon or its underlying dimensions or characteristics [14]. Descriptive theories are the most basic types of theories, because 'describing what is' does not explain causality or provide predictive generalizations [13].

There are a number of criteria we can use to evaluate the quality of any typology. Figure 2 visualizes the four criteria we considered most important for the specific need typology we were developing (based on [49,50]). The rectangles represent the typology as a whole, and the grey ellipses represent the types or categories that are demarcated in the typology. Rectangle A (top row) shows a representation of higher-quality typologies; rectangles B, C and D (middle row) show representations of lower-quality typologies; rectangles E and F (bottom row) show representations of typologies with low versus high granularity.

The first three criteria—inclusion, distinction, and equivalence—test the theoretical soundness of a typology. These criteria evaluate whether types cover the entire phenomenon, overlap as little as possible, and exist on the same level of abstraction. The evaluation of the fourth criterion, granularity, depends heavily on the intended application of the typology. Imagine creating a typology of drinkware. For a waiter at a restaurant, a useful typology should distinguish glasses and cups according to their function and etiquette of use, and probably includes dozens of categories, such as red wine glasses, teacups, espresso cups, and so forth. For a recycling plant, a useful typology of drinkware would distinguish glasses and cups according to the extent to which they can be recycled together, and include a few categories such as ceramic drinkware, clear glassware and coloured glassware. The first typology cannot be called better simply because it contains more categories, since for its application, the categories of the second typology suffice. Nevertheless, for two typologies that serve the same purpose and application, the one with higher granularity will provide more information and utility.

The Hierarchy of Needs introduced some important ideas to the development of motivational theory, but it is not optimal as a resource of fundamental needs for the design discipline. The main reason is that its granularity is too low. The five need categories are meta-types instead of types. Design is better informed by need types than by generic need categories. In design practice and research, this is often resolved by resorting to examples of needs (see the right side of Figure 1). These examples, however, are formulated on the object level. As a consequence, they are not necessarily universal and are somewhat arbitrary.

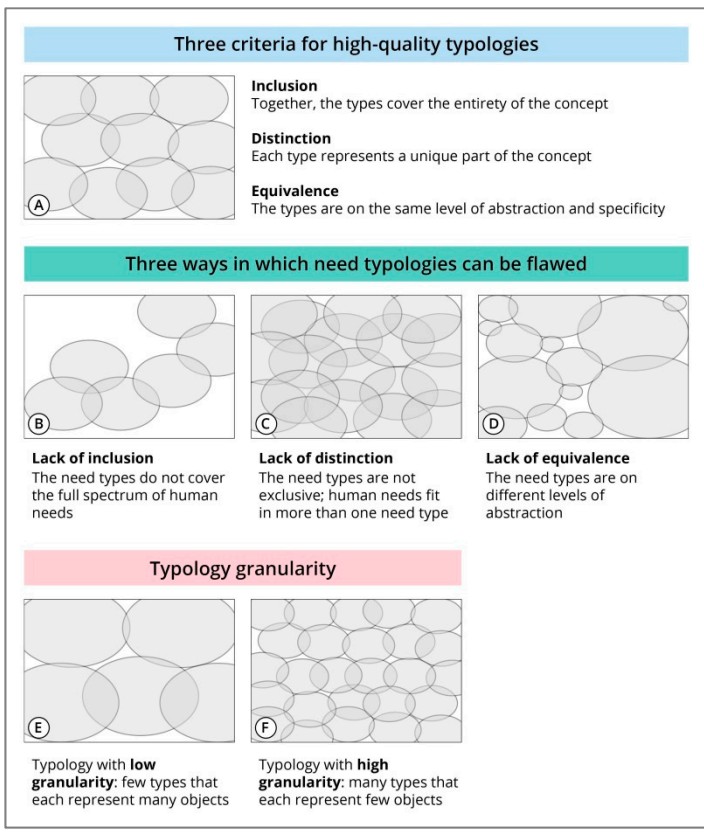

**Figure 2.** Four criteria for evaluating typologies.

The new typology was developed to overcome these limitations by providing a more granular overview of fundamental need types. A key challenge in the process was to distinguish needs that are fundamental from those that are not. Many authors of published need typologies did not explain on what basis they decided if a need is fundamental or not, and those who did often used different criteria. For our review, we adopted the approach of Baumeister and Leary, who provided the most rigorous operationalization and the most detailed set of criteria [51]. They proposed the five requirements for fundamental needs, presented in Table 2.

**Table 2.** Five criteria for fundamental psychological needs.

| A need is fundamental when it ... |
| --- |
| (1) is universal, that is, it applies to all people, transcending cultural boundaries; |
| (2) is not derived from another need; |
| (3) leads to (physical or mental) well-being (that goes beyond momentary pleasure) when fulfilled, and to pathology (medical, psychological, or behaviour) when unsatisfied; |
| (4) motivates behaviour in a wide variety of situations (not only in specific circumstances); |
| (5) affects a wide and diverse assortment of behaviours. |

The new typology was developed by collecting fundamental needs from a variety of psychological theories. We found the following six typologies of fundamental needs to meet the five criteria: Deci and Ryan's Self Determination Theory [16,40,52,53], Sheldon et al.'s factors of well-being [5], Ford and Nichols' taxonomy of fundamental human goals [21,54], Ryff's determinates of psychological well-being [39,55], and the human values typologies created by Schwartz [18,56] and Rokeach [20,57]. In addition, we studied the collected works of Maslow [25,31,32,58] to honour the depth of his original ideas in our analysis. To give an example, in his later work, Maslow [31,58] proposed

several more needs, including Cognitive needs (knowledge and understanding, curiosity, exploration, need for meaning and predictability), Aesthetic needs (appreciation and search for beauty, balance, form), and Transcendence needs (mystical experiences and certain experiences with nature, aesthetic experiences, sexual experiences, service to others, the pursuit of science, religious faith). We used a semi-systematic (narrative) review approach with a form of thematic analysis, following the six-phase approach of Braun and Clarke [59] (pp. 86–93). First, we (the two authors) familiarized ourselves with the typologies by reading the manuscripts and focusing on the fundamental needs, definitions, explanations, examples, and supporting data they propose (phase 1), and coded the need concepts we encountered (phase 2). Next, we created a first set of fundamental needs as initial themes or candidates (phase 3). We reviewed candidates, determined which were sufficiently supported by the data, and which could be collapsed into each other—two seemingly separate needs might form one need—according to the criteria in Table 2 (phase 4). The next phase involved several rounds of re-reading and discussion sessions to refine the themes into the final set of fundamental needs (phase 5). Lastly, we produced and collected examples, images, and designs that represent the different needs (phase 6). Any differences in interpretations between the authors were resolved in discussion sessions in which they would argue for their position with literature and examples until a consensus was reached. Oftentimes, these consensuses were considered better solutions than either of the initial positions from which they had emerged.

Once a satisfactory need overview was developed, we re-examined the literature to develop coherent definitions. This process resulted in the final overview of thirteen fundamental needs in Table 3. An overview of included need concepts can be found in Appendix A.

**Table 3.** Thirteen fundamental psychological needs and 52 sub-needs.

| Fundamental Needs | Explanation | Sub-Needs |
|---|---|---|
| Autonomy | Being the cause of your actions and feeling that you can do things your own way, rather than feeling as though external conditions and other people determine your actions. | - Freedom of decision<br>- Individuality<br>- Creative expression<br>- Self-reliance |
| Beauty | Feeling that the world is a place of elegance, coherence and harmony, rather than feeling that the world is disharmonious, unappealing or ugly. | - Unity and order<br>- Elegance and finesse<br>- Artistic experiences<br>- Natural beauty |
| Comfort | Having an easy, simple, relaxing life, rather than experiencing strain, difficulty or overstimulation. | - Peace of mind<br>- Convenience<br>- Simplicity<br>- Overview and structure |
| Community | Being part of and accepted by a social group or entity that is important to you, rather than feeling you do not belong anywhere and have no social structure to rely on. | - Social harmony<br>- Affiliation and group identity<br>- Rooting (tradition, culture)<br>- Conformity (fitting in) |
| Competence | Having control over your environment and being able to exercise your skills to master challenges, rather than feeling that you are incompetent or ineffective. | - Knowledge and understanding<br>- Challenge<br>- Environmental control<br>- Skill progression |

**Table 3.** *Cont.*

| Fundamental Needs | Explanation | Sub-Needs |
|---|---|---|
| Fitness | Having and using a body that is strong, healthy, and full of energy, rather than having a body that feels ill, weak, or listless. | - Nourishment<br>- Health<br>- Energy and strength<br>- Hygiene |
| Impact | Seeing that your actions or ideas have an impact on the world and contribute to something, rather than seeing that you have no influence and do not contribute to anything. | - Influence<br>- Contribution<br>- To Build something<br>- Legacy |
| Morality | Feeling that the world is a moral place and being able to act in line with your personal values, rather than feeling that the world is immoral and your actions conflict with your values. | - Have guiding principles<br>- Acting virtuously<br>- A just society<br>- Fulfilling duties |
| Purpose | Having a clear sense of what makes your life meaningful and valuable, instead of lacking direction, significance or meaning in your life. | - Life goals and direction<br>- Meaningful activity<br>- Personal growth<br>- Spirituality |
| Recognition | Getting appreciation for what you do and respect for who you are, instead of being disrespected, underappreciated or ignored. | - Appreciation<br>- Respect<br>- Status and prestige<br>- Popularity |
| Relatedness | Having warm, mutual, trusting relationships with people who you care about, rather than feeling isolated or unable to make personal connections. | - Love and intimacy<br>- Camaraderie<br>- To nurture and care<br>- Emotional support |
| Security | Feeling that your conditions and environment keep you safe from harm and threats, rather than feeling that the world is dangerous, risky or a place of uncertainty. | - Physical safety<br>- Financial security<br>- Social stability<br>- Conservation |
| Stimulation | Being mentally and physically stimulated by novel, varied, and relevant impulses and stimuli, rather than feeling bored, indifferent or apathetic. | - Novelty<br>- Variation<br>- Play<br>- Bodily pleasure |

As a final step in the typology development, we consulted the (non-fundamental) need typologies of Wicker et al. [23], Talevich et al. [22], Chulef et al. [24] to identify good examples of sub-needs. This resulted in the 52 sub-needs featured in the third column of Table 3. The main purpose of listing these sub-needs is to offer a more in-depth illustration of the fundamental needs. They were carefully selected to show the range and scope of the fundamental needs, while still representing relatively universal and general concepts. However, the sub-needs are themselves not fundamental, as they do not meet all the criteria in Table 2. Most notably, they fail to meet criterion 2, and to a lesser extent criterion 1. For instance, the need for 'financial security' is an operationalization of the need for security that holds true for many, but not all people. The sub-needs can, in turn, take shape as and through goals and desires that are specific to particular individuals and/or situations. The sub-need for camaraderie,

for example, can manifest in the goal to spend weekends with friends, the desire to develop a friendly relationship with colleagues, the intention to meet new people at a party, et cetera.

## 5. Design Opportunities

The fulfilment of psychological needs provides pleasure and contributes to long-term well-being [5]. In fact, any positive experience ultimately stems from the fulfilment of some psychological need [8]. Need fulfilment is therefore a major source of positive user experiences with technology, and these experiences can be categorized on the basis of the primary psychological needs they fulfil [6,17,60]. This applies to user experiences across life domains, including those in leisure and work activities [61]. Moreover, because they address the content of these experiences, psychological needs can be used as a basis for categorizing and distinguishing positive user experiences [62]. A needs typology provides designers with a lexicon of psychological needs they can use when communicating with clients, within design teams and with end-users and other stakeholders. A commonly-known vocabulary can be used to operationalize the well-being impact of existing or new designs.

There are two main ways a product can interact with our needs: the product can fulfil needs (profile 1), and the product can harm needs (profile 2). Profile 1 represents sources of positive experiences with a product, and profile 2 represents sources of negative experiences. Some products can fulfil a single need, like an energy bar (fitness), a water gun (stimulation) or a lock (security). Most products, however, fulfil multiple needs, in which case the needs can be ranked by order of significance. For example, in-ear headphones fulfil the need for stimulation, comfort and autonomy. The use of a product can also harm needs. The fulfilment of one need might actually detract from the fulfilment of another. For example, high-heeled shoes are considered elegant footwear in many cultures, but are also a source of impracticality (harming the need for competence) and physical displeasure (harming the need for comfort). Need profiles are an excellent starting point to use fundamental needs as inspiration for new product ideas. We see at least three design opportunities: (1) strengthen current needs; (2) introduce new needs; and (3) reduce harmed needs.

### 5.1. Design Opportunity 1: Strengthen Current Needs

Fundamental needs are a powerful resource for artefact redesign and ideation. Because the set of fundamental human needs represents a host of fundamental drivers of human behaviour, it offers designers insight into (nearly) everything people might want to derive from product experience. The first design opportunity is to strengthen the needs that are already fulfilled by a product or service. This is a useful opportunity if you want to increase the relevance of and user engagement with a product without changing its essence or core functionality. The first step in approaching this opportunity is to compile a product-specific needs profile. What needs does the product fulfil? The design effort can focus on creating additional ways for the product to fulfil these needs. They might consider different aspects of the product, or the different roles it could play in users' lives. Another strategy is to work with sub-needs: identify a particular sub-need that the product satisfies and find features or solutions that could fulfil other sub-needs. An example of a design that was based on this strategy is the Journey to Yourself fitness tracker application by designer Karen Gonzalez ([63]; see Figure 3, left). She redesigned an activity tracking application co-creatively with product users. The original design fulfilled the need for fitness by focusing on the sub-need of physical exercise. The redesign broadened the scope by also fulfilling other fitness sub-needs, such as the need for good nutrition and the need for mental health.

| Journey to yourself | Good-Bag | Wheelchair for children |
|---|---|---|
| Design by Karen Gonzalez | Design by Simon Akkaya | Design by Eva Dijkhuis |
| 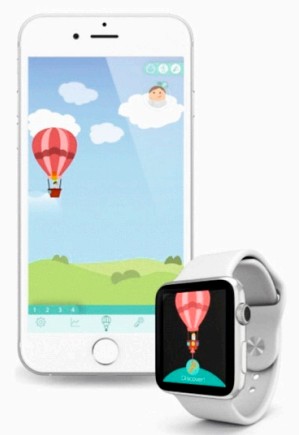 | 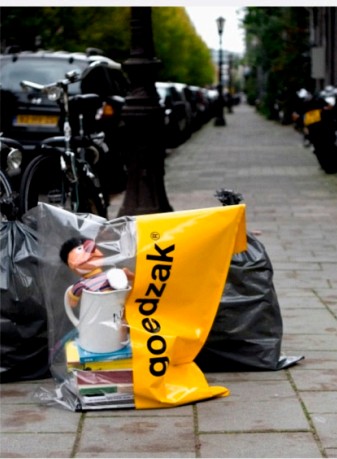 | 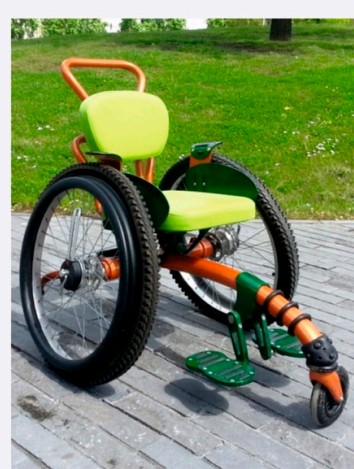 |
| The *Journey to Yourself* is an activity tracker that uses a different narrative than a regular sports coach app. Your journey to a healthy lifestyle is visualized as a balloon ride. You start your journey by choosing healthy habits in the domains of nutrition, physical activities and mental activities. Activity completion translates to the winds that propel the balloon. Occasional downtime is not a negative but a natural part of your journey. | The *Good-Bag* is a highly-visible plastic bag you can use to give your unwanted products a chance at a second life: fill it with the items you no longer need and put it on the street. Anyone can pick out anything they value. The new owner will feel good about your altruistic act, and you will contribute to sustainability. The bright yellow panel attracts attention, and the transparent part makes the content visible to passers-by. | The wheelchair for children focuses on the autonomy of the user. Most conventional wheelchairs have handles that express the dependency of the children; it literally makes their wheelchair a 'push-chair'. For the new design, the handles were removed. Instead, it has a push bar, which the child can slide down behind the back seat, where it becomes unrecognizable as a push bar. |

**Figure 3.** Design cases informed by needs analysis.

### 5.2. Design Opportunity 2: Introduce New Needs

Designer Simon Akkaya redesigned the everyday garbage bag with one particular fundamental need in mind: the need for morality. The result was the Good-Bag (Figure 3, middle), which enables the user to donate unwanted possessions that are still in good shape for reuse (see [64] for a description). Many products have become synonymous with their core needs—microwaves heat food, pillows give physical comfort and insurance policies provide security. But why should products be limited to the needs they were originally created to fulfil? Perhaps a couch could be mentally stimulating, or a microwave could make the user feel more appreciated, or a plastic bag could be used to act on one's values? If a product is (re)designed to satisfy needs that were not listed in its existing needs profile, it will likely become (even) more relevant to users. In this design opportunity, the overview of fundamental needs becomes a source of creative inspiration. Each need can be the starting point for exploration of new product features and functionality. The Good-Bag fulfils the need for morality by lowering the threshold for people to engage in acts of altruism: donating their possessions to anonymous passers-by.

### 5.3. Design Opportunity 3: Reduce Need Harm

The third design opportunity focuses on needs that are harmed by the product. The first step is to identify what needs the product harms, which may or may not be unintentional. Next, the product can be redesigned to resolve these harmed needs. The wheelchair for children by Eva Dijkhuis (Figure 3, right) was based on this strategy. With a user study, she determined what needs were harmed by conventional wheelchairs. She found that one of the main shortcomings of conventional models was

that they harm the need for autonomy [65]. The core of the problem was in the handles that are used to push the wheelchair. While a wheelchair offers mobility and freedom to move, the design also expresses dependence, because the prominent handles signal that the person can, and needs to be, pushed around. The wheelchair redesign was intended to create a new form that removes the association of dependence.

## 6. A Series of Typology Communications

We explored which means would best communicate our typology of fundamental needs to design researchers and practitioners. We ended up developing a number of different examples, based on the insight that knowledge on multiple levels of abstraction increases the extent to which it is actionable to designers [66]. For each means, we used both verbal and visual communication, as the combination fosters deeper understanding and learning [67]. Design method cards often use both text and images to illustrate ideas and principles, because the images make the principles less abstract and enable people to intuitively relate to concepts [68–70]. We developed three means for communicating the typology, which are briefly described below: (1) an illustrated needs overview, (2) a poster entitled '13 chairs—13 needs', and (3) a portfolio of alarm clocks.

### 6.1. The Illustrated Needs Overview

The first means is an illustrated needs overview (Figure 4). Each need is illustrated by a representative image and a short definition. The images were selected over the course of a three-stage procedure (for a detailed report, see [71]). First, two researchers collected 260 images (20 for each need) from a variety of online image databases. That set was then reduced to 10 images per need by a team of three design researchers. Three criteria guided the selection: the image (1) expresses the target need, (2) does not also express another need, and (3) adds diversity to the overall set. In the third step, a questionnaire study (N = 72) examined the degree to which each of the 130 images was able to express the target need. Respondents rated all images one a 7-point Likert scale (1 = 'the image does not represent this need at all'; 7 = 'the image is an excellent representation of this need'). The results were used to select the final images for the illustrated need overview. Depending on the context of application, the overview can be used as an A4 sheet, or as a set of thirteen cards.

### 6.2. The '13 Chairs—13 Needs' Poster

The second communication means is an overview of thirteen chairs (Figure 5). Each chair represents one of the fundamental needs. The designs were selected by the authors using online image databases. In our search, we looked for designs in which (a) the designers explicitly expressed an intention to address a target need (e.g., in catalogue descriptions or interviews), and (b) the design (in our opinion) convincingly expresses/communicates that target need. The main aim of this poster is to illustrate that the overview of needs can serve as a means for inspiration. The poster demonstrates that even a simple product like a chair can, in principle, address every need, albeit that some needs are a bit more exotic to the use of chairs than others.

### 6.3. The Inspiration Card Deck

The third tool is a deck of 16 inspiration cards called 'Wake Up and Smell the Coffee' [72,73]. Figure 6 shows some examples. The cards portray conceptual alarm clock designs and corresponding fundamental need profiles. The alarm clocks were designed in a previous design project, which aimed to provide pleasurable and meaningful wake up experiences that increase well-being. A questionnaire study (N = 34) examined the degree to which the designs fulfil the fundamental needs. For each design, respondents were asked to allocate ten points across the thirteen needs. This resulted in cards with need profiles visualized as bar graphs. The cards are intended to be used in an educational setting, to support a discussion about product design and needs fulfilment. The front of each card depicts an alarm clock design with a short description; the back of each card shows the needs profile. Bars of

needs with an average rating of 1.0 or higher are highlighted. We decided to place the needs profiles on the back to support the educational card usage. Students are first given the cards (and asked not to look at the back side) and invited to envision which fundamental need(s) the designs might fulfil. Next, they can look on the back of the card to compare their vision with the needs profile. This procedure serves as a conversation starter for a discussion about fundamental needs in relation to design.

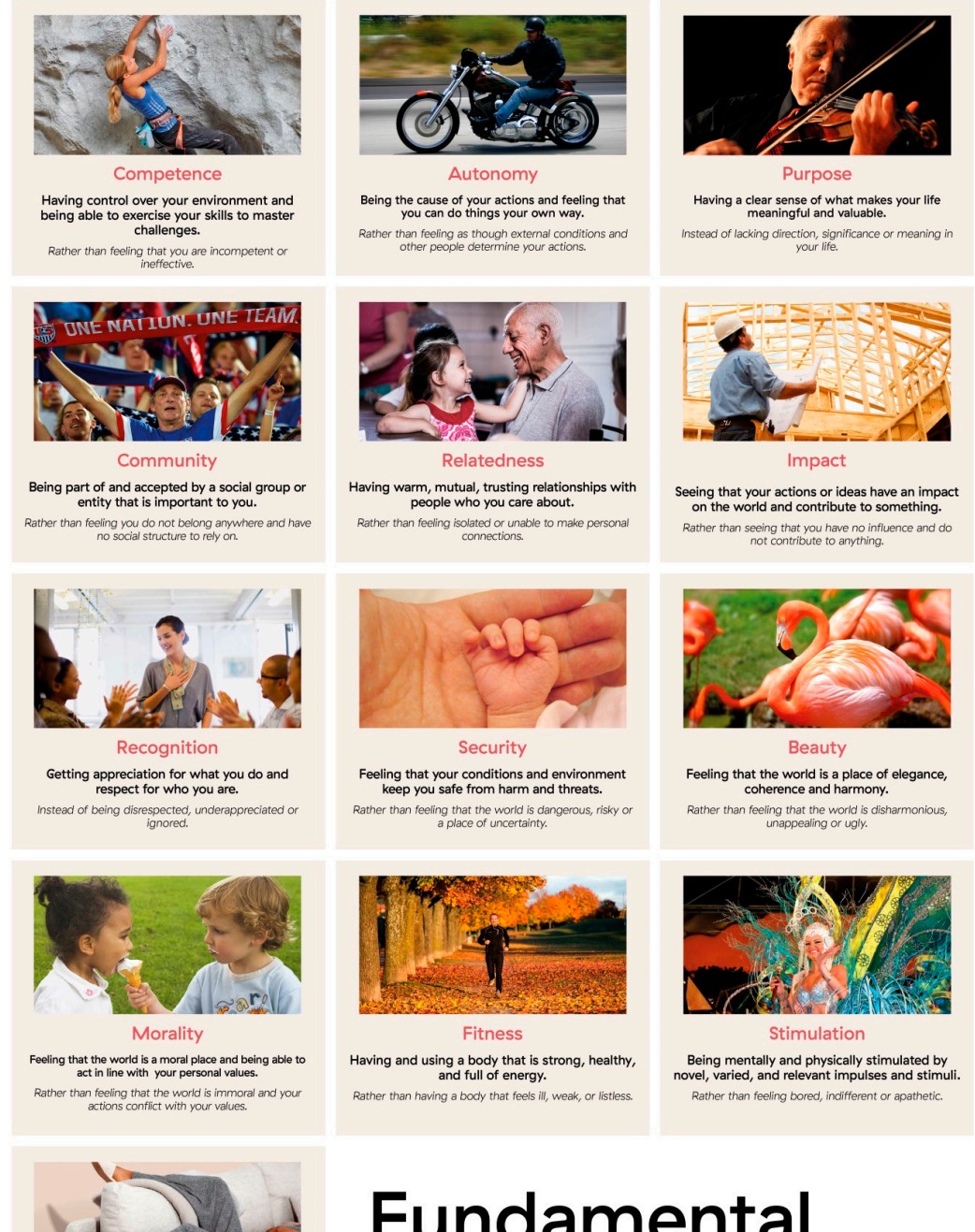

**Figure 4.** Illustrated overview of the fundamental needs typology.

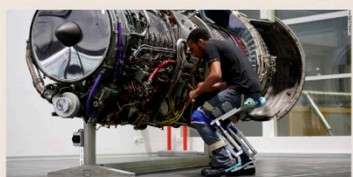

### Competence

Some jobs require constant switching between moving around and sitting down. With the Chairless chair, you can sit wherever, and whenever you want, enabling you to work efficiently and comfortably.

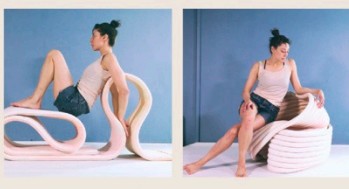

### Autonomy

The Body-Morphing chair is a six-meter long, rice-filled cushion. You can explore how you want to sit and then shape the chair to match your preferred body posture at that moment.

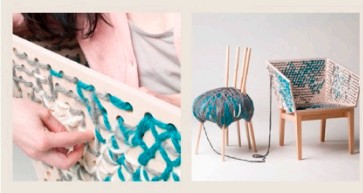

### Purpose

Stitch and Wooly are half-finished. You have to complete them yourself through stitching and embroidery. If you commit to the time-intensive process, you will be awarded with a chair that is unique and personal.

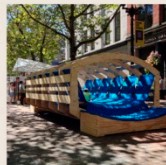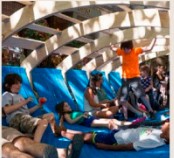

### Community

The woven rocker is a five-meter long, outdoor rocking chair for large groups of people. The textile sitting surface gently passes on the movements of every person, making you feel connected to the group.

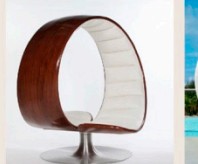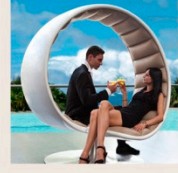

### Relatedness

The Hug chair comfortably fits two people. Meet someone new or enjoy a heart-to-heart moment with your loved-one while relishing the luxurious aesthetics of this shared bubble.

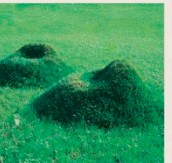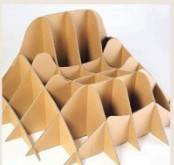

### Impact

Terra! is an outdoor chair that literally needs time to grow. Cover a cardboard construction with soil and grass seeds and then allow nature to take over. With some patience, you create a landmark to sit on.

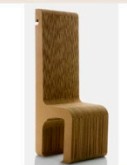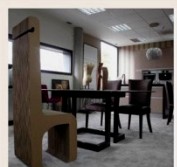

### Recognition

Royalty is familiar with the principle: the shape of a seat can enhance your stature. The proportions of the Valentina Throne chair give you that royal dignity: you are a king or queen on your throne.

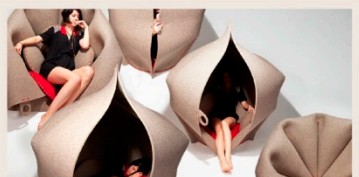

### Security

The Hush pod is a soft chair that can shield you from the outside world. Wrap the felt chair around yourself to create a cocoon-like safe space, giving you comfort, privacy and protection.

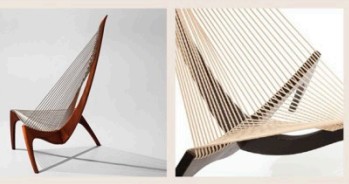

### Beauty

The iconic Harp Chair is a sculptural object that was inspired by a Viking ship's bow section. The woven flag halyard gives the design an intriguing optical quality.

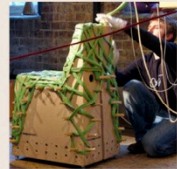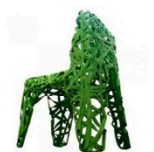

### Morality

The handwoven RD4 chair is completely made out of recycled waste plastic. You can feel content about your modest but vital contribution to reducing the pile of plastic that pollutes the oceans.

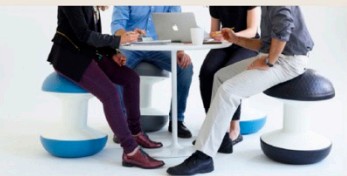

### Fitness

Yes, conventional office chairs are comfortable, but they also promote inertia. The shape of the Ballo instead requires you to constantly use your core muscles to sit upright, helping you stay fit at work.

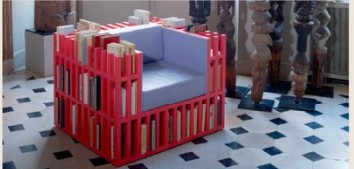

### Stimulation

The Bibliochaise is a comfortable armchair equipped to hold up to 300 books, keeping them all within arm's reach. You can park yourself and read for hours without having to leave your spot.

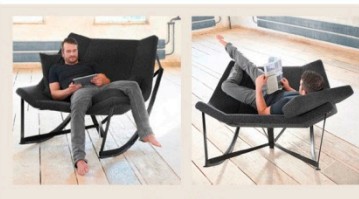

### Comfort

The Sway Rocking chair is a hide-out from the daily grind. Find your favoured position, surrender to the gentle flow of movements and feel like you're floating on a calm ocean. Life can be so relaxing.

- Chairless chair by Noonee
- Body-Morphing chair by Kirsi Enkovaara
- Stitch and Wooly by Susanne Westphal
- Woven Rocker by Tarboo and LMN Architects
- Hug chair by Gabriella Asztalos
- Terra! by Studio Nucleo
- Valentina Throne by Sanserif Creatius
- Hush Pod by Freyja Sewell
- Desile folding chair by Christian Desile
- RD4 chair by Cohda Design
- Ballo by Chadwick Studio
- Bibliochaise by Alisée Matta and Giovanni Gennari
- Sway Rocking chair by Markus-Krauss
- Vintage Dining chair by Hugues Revuelta

## Thirteen chairs, thirteen fundamental needs

The chair – one of the most basic pieces of furniture. The prototype has four legs, a seat and a back. It holds your weight and supports an upright position for an extended period of time. But that is just the beginning: Chairs have been designed to serve countless additional purposes. From giving us a moment of privacy to helping us to connect. There is a chair for every need!

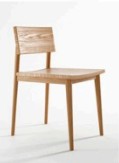

**Figure 5.** Poster with thirteen chairs representing thirteen fundamental needs.

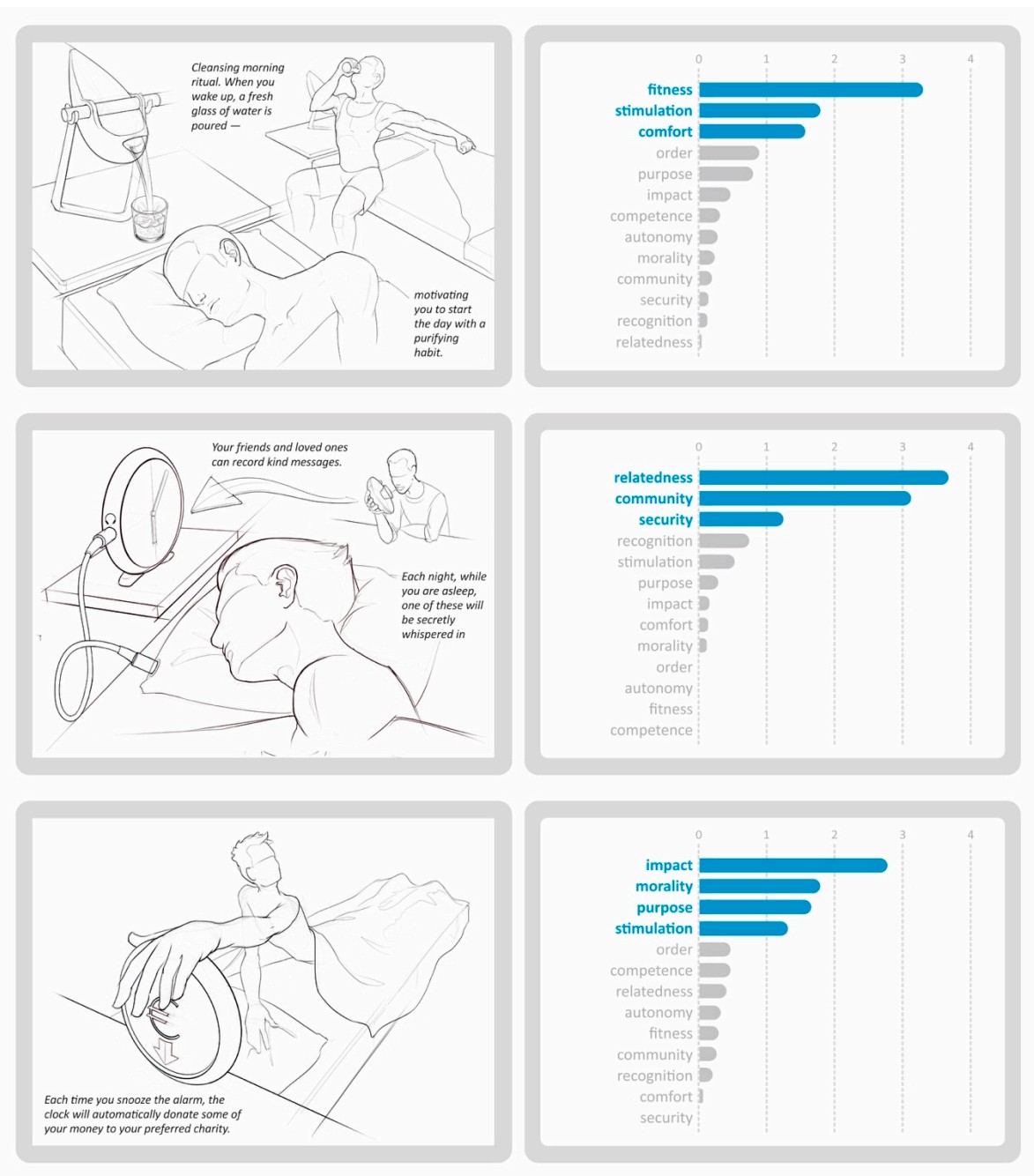

**Figure 6.** Three examples of cards from the 'Wake up and smell the coffee' card deck (left = front; right = back).

## 7. Discussion

This paper introduced a design-focused typology of psychological human needs with thirteen fundamental needs and 52 sub-needs. Typologies help us to "make sense of the world"—they enable us to understand and analyses complex and otherwise disorderly concepts [15]. The purpose of the fundamental need typology is to serve as a need repertoire to design practice and design research. The application value is pragmatic: Needs can be drawn from it when formulating need profiles to represent what truly matters to end-users. Höök & Löwgren [74] and Löwgren [75] proposed that there is a space between formal theory on the one hand, and a design prototype on the other hand. They named this the space of 'intermediate-level knowledge'. Examples are design guidelines and

design patterns. By using illustrations and examples, the three communication means (the illustrated overview, the poster, and the cards in Figures 4–6), fit in this space of intermediate-level knowledge.

Since we started developing the typology nearly ten years ago, we have tested several iterations in both academic and commercial contexts. We found that students, designers, and clients alike can be enlightened and inspired by a concise yet exhaustive overview of what people need in life. Numerous student and commercial projects confirmed the merit and broad applicability of the typology, as it has been applied in domains ranging from elderly care to financial advice tools, and from air travel to well-being interventions. The typology can be used in when studying the needs of people in a specific context (e.g., in an airport or at the gym) or of people who are engaged in a specific activity (e.g., providing healthcare or teaching children). In those cases, it can function as an inclusive framework to cluster and make sense of the hundreds of context-specific needs captured in a study. Note that the need typology should be used with an awareness that it reduces the infinite complexity of the human biological, psychological, and sociocultural systems. This applies to all typologies of human qualities—it is the price to pay for a reductionism-based interpretation of people. The fundamental need typology is a lens that influences what we see and thus which design opportunities we identify. Moreover, an individual's repertoire of goals, values, drives and other motivations are influenced rather than determined by their fundamental needs [36]. Life-span theories of motivation propose that the salience of specific needs change as one moves through life (for a review, see [76]). As people grow older, they become increasingly aware that time is limited, and this perception of time is closely linked to the pursuit of need fulfilment and the subsequent impact on their well-being ([77–79]). With increasing age comes an appreciation of the fragility of life, and a person's wellbeing is influenced by their ability to be effective in adapting their goals to their lifespan development [80]. This illustrates that, while the fundamental needs form the bedrock of any person's behavioural repertoire, this repertoire is ultimately unique for any individual as it is self-constructed by experience, socialization, and life-long development [2]. In other words, when using the typology of fundamental needs, we should be alert that, like any reductionist theory, it represents but a corner of the space of opportunities to design for meaningful contributions to the dynamic, and ever-evolving holistic reality of any human being.

With our project, we aimed to acknowledge the value and impact of Abraham Maslow's motivational theory, while at the same time highlighting that its application should go hand-in-hand with an understanding of its limitations. In our experience, most design students and even some researchers are not aware that the Hierarchy of Needs can only be fully understood in relation to the historical, scientific, and societal context in which it was developed. Even so, Maslow was a brilliant thinker, and his scientific legacy is immensely richer and more nuanced than his best-known work, the Hierarchy of Needs. When reading a book like "Motivation and personality", one can only be amazed to see how much of Maslow's original thinking is echoed in the contemporary Positive Psychology movement and the related Positive Design and Positive Technology movements.

One intriguing limitation of Maslow's motivational theory is its heavy focus on the individual. In their global study, Tay and Diener found that a person's subjective well-being is not only determined by the extent to which their individual needs are fulfilled, but also by the extent to which the needs of other people and society are fulfilled [4]. This finding aligns with Gambrel and Cianci's literature review [81]. They found that the most basic need in collectivist cultures is the need for community, and that in these cultures a person's individual development (self-actualization) is attained through societal development. These findings suggest that improving life for the individual goes hand in hand with improving society. Moreover, social scientists have shown that well-being in individualistic cultures also include social dimensions. The influential social model of well-being of Keyes [46] proposes that an individual's well-being is partly determined by their appraisal of how they function in society, with five dimensions of social well-being: Social integration (being part of society), social contribution (contributing to society), social coherence (making sense of society), social actualization (believing in the positive evolution of society), and social acceptance (holding favourable views of fellow citizens). In our typology, social well-being is represented by the needs for community, morality,

and impact. While we believe that these three fundamental needs represent the social dimensions at a general level, future research can examine if they fully capture the social factor of well-being and which (additional) sub-needs can help to operationalize the social perspective in design applications. In addition, a possible next step may explore how the needs typology can support social design. This idea aligns with the observation that social design thinking is often based on the premise that design should address fundamental needs [82]. Victor Papanek, whose quote opened this paper, was one of the first to address issues of social design. He wanted to change the design field by rejecting design that does not account for the needs of all people and that disregards environmental consequences.

A final reflection focuses on the value of balanced needs. Well-being psychologists have found that all fundamental needs ought to be satisfied to a certain degree for a person to be happy, regardless of whether this person consciously values these needs. Fulfilling the thirteen needs makes a distinct contribution to a person's subjective well-being. It is not possible to substitute one need with another. For example, if you realize you lack personal connections (relatedness), you can try to compensate by focusing more attention on work and skills (competence), but that will not be effective in the long run. Psychologists are increasingly understanding the importance of balanced needs fulfilment. Sirgy and Wu [83], for example, proposed that the 'balanced life' is, all things being equal, a more desirable one. Diener and colleagues [84] supported this hypothesis with the finding that a need's fulfilment demonstrates declining marginal utility. Thus, because people need to fulfil a variety of needs, it is likely that a mix of daily activities that includes mastery, social relationships, and the meeting of physical needs is required for optimal subjective well-being. This has an important implication for the design for well-being: the contribution of a need-fulfilling design intervention to the overall well-being of an individual can only be understood in proportion to that person's overall need-fulfilling capabilities and opportunities in all of his or her activities, including those that may not be related to the (activity of using the) design at hand, but are not less relevant to that person's self-actualization.

**Author Contributions:** Both authors made an equal contribution to this manuscript, including the conceptualization, methodology, analysis and writing. All authors have read and agreed to the published version of the manuscript.

**Funding:** This research was supported by VICI grant number 453-16-009 of the Netherlands Organization for Scientific Research (NWO), Division for the Social and Behavioural Sciences, awarded to P.M.A. Desmet.

**Acknowledgments:** Jort Nijhuis made the drawings of the alarm clock designs in Figure 6. Petra Salaric and Veerle van Engen made the graphic design of Figures 4 and 5. The Journey to Yourself (Karen Gonzalez), Good-Bag (Simon Akkaya) and Wheelchair for Children (Eva Dijkhuis) were designed by students of the Faculty of Industrial Design Engineering of Delft University of Technology. The pyramid image in Figure 1 was created by SunnyWS (Shutterstock); words in Figure 1 were added by the authors. We extend our gratitude to the anonymous peer reviewers for their helpful comments on an earlier draft of the manuscript.

**Conflicts of Interest:** The authors declare no conflict of interest. The funders had no role in the design of the study; in the collection, analyses, or interpretation of data; in the writing of the manuscript, or in the decision to publish the results.

## Appendix A. Overview of Needs from the Fundamental Need Typologies

Table A1 gives an overview of the six typologies of fundamental needs that served as the basis for the typology of 13 psychological needs (for references, see the first row in Table A1): Deci and Ryan's Self Determination Theory, Sheldon et al.'s factors of well-being, Ford and Nichols' taxonomy of fundamental human goals, Ryff's determinates of psychological well-being, and the human values typologies created by Schwartz and Rokeach.

**Table A1.** Five sources of fundamental psychological needs.

| Fundamental Needs | Deci & Rian [16,40,52,53] | Sheldon, Elliot, Kim, & Kasser [5] | Ryff & Keyes [39,55] | Ford & Nichols [21,54] | Schwartz [18,56] | Rokeach [20,57] |
|---|---|---|---|---|---|---|
| Autonomy | Autonomy | Autonomy | Autonomy | Individuality; Self-determination; Creativity | Self-direction; Independence | Freedom |
| Beauty | | | | | A world of beauty | Beauty |
| Comfort | | | | Tranquillity | Inner harmony | Inner Harmony; Peace; Comfortable life |
| Community | | | | Belongingness; Unity | Tradition; Honouring elders; Obedience | |
| Competence | Competence | Competence | Environmental mastery | Mastery; Understanding; Exploration | Mastery; Being capable; Control | |
| Fitness | | Physical thriving | | Physical well-being | | |
| Impact | | Influence | | Resource provision | Having influence; Ambition; Being successful | Accomplishment |
| Morality | | | | Equity; Social responsibility | Self-discipline | Equity |
| Purpose | | Self-actualization; Meaning | Purpose in life; Personal growth | Transcendence | Spiritual life | Salvation |
| Recognition | | Popularity | | Resource acquisition; Superiority | Social approval; Authority; Social power | Social recognition |
| Relatedness | Relatedness | Relatedness | Positive relations | | True friendship; Mature love | Mature love; True friendship |
| Security | | Security | | Safety | Security | Family security; National security |
| Stimulation | | Pleasure/Stimulation | | Bodily sensations; Entertainment | Variety; Pleasure | Exciting life; Pleasure |

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
