# Peer review of "Beyond Maslow’s Pyramid: Introducing a Typology of Thirteen Fundamental Needs for Human-Centered Design"

_mti, doi:10.3390/mti4030038_

Round 1

Reviewer 1 Report

The authors developed a need-centered framework to  be used for design for wellbeing. In the design for positive user experience and design for wellbeing, need frameworks were an important step forward for research in practical application (Calvo & Peters, 2014; Hassenzahl, 2008; Hassenzahl et al., 2010; Tuch et al., 2016). This paper builds on this central idea and it takes a step forward. The contribution of the presented framework is, that it is systematically developed, based on a review of seven need theories, and provides a differentiation of fundamental needs and sub-needs. Since need-centered design is important for design research and practical design, a differentiated and well-founded need framework is an important step forward for the HCI community.

The authors started with the Maslow’s seminal work on hierarchy of needs. This framework has been used in research and application in heterogenous disciplines. The authors analyzed and evaluated Maslow’s framework in detail. Especially, the considerable critique of the framework has been valued in detail, which is necessary and useful. In a second step, a systematic extension based on six further need models and a rule-guided definition of fundamental needs and sub-needs was made. Finally, as a third step, the framework was prepared as tools to be used for design practice. Even the process of creating these tools was founded by empirical studies.

The paper is well written and it is easy to follow the argumentation.

There are some minor improvements necessary:

  1. Sentence in line 43 – 45 needs a little bit more explanation

The sentence “Designs do more than serve the needs of end users — they embody the needs of every single person (including the designers) involved in their ideation, development, execution and distribution.” should be explained a bit more. It sounds interesting and relevant, but should be understandable to readers who are not deeply involved in practice of design for wellbeing as well.

  1. Differentiation necessary in line 100-101

The following sentence needs a bit more differentiated view: “Moreover, any typology of needs is neither true nor correct as such, because its usefulness depends on the purpose of its application [13].” In a scientific sense a typology should be true. For example, a typology with non-valid needs would be useless. But I agree, that the detailedness and structure of a typology could depend on its usage.

  1. Chapter “4. Developing an Improved Typology of Fundamental Needs”

The process of extending Maslow’s typology based on six further need models is the kernel of the development of the extended framework. The method of doing so is described in just one paragraph (line 329-339). Here some more information on the method and process is necessary:

  • The guidelines of Brown and Clark and the way they have been applied should be described in some sentences.
  • How did the authors cope with differences in interpretation? How big were the differences between the interpretations of the authors? This is relevant, because needs, having the same title in different typologies, are not described in the same way already by the typology authors. How was this solved?
  • Appendix A is very helpful, but it would be more interesting to see the source typology of the needs in table 3. Maybe appendix A could be improved by incorporating table 3 and adding the source typology.
  1. Grammatic problem in line 513

It seems to be that the sentence in line 513 has a grammatic problem. It is hard to understand.

  1. Additional source for line 357 – 360

I agree with the conclusion of the two sentences of line 357 – 360. To make the point stronger it might be helpful to cite Tuch et al. (2016) who tested the Hassenzahl model in leisure and work contexts.

Literature

Calvo, R. A., & Peters, D. (2014). Positive Computing - Technology for Wellbeing and Human Potential. MIT Press.

Hassenzahl, M. (2008). User experience (UX): towards an experiential perspective on product quality. Proceedings of the 20th International Conference of the Association Francophone d’Interaction Homme-Machine, 11–15. http://portal.acm.org/citation.cfm?id=1512717

Hassenzahl, M., Diefenbach, S., & Göritz, A. (2010). Needs, affect, and interactive products – Facets of user experience. Interacting with Computers, 22(5), 353–362. https://doi.org/10.1016/j.intcom.2010.04.002

Tuch, A. N., van Schaik, P., & Hornbæk, K. (2016). Leisure and Work, Good and Bad: The Role of Activity Domain and Valence in Modeling User Experience. ACM Transactions on Computer-Human Interaction (TOCHI), 23(6), 35. https://doi.org/10.1145/2994147

Author Response

We thank the reviewers for their careful reading of our manuscript and their many insightful comments and suggestions. Below we respond to the comments of Reviewer 1 in detail, with reviewer comments in bold. Note that the line numbers in our responses refer to the lines in the revised manuscript (with track changes), which may differ from the original manuscript. 

There are some minor improvements necessary:

1) Sentence in line 43 – 45 needs a little bit more explanation

The sentence “Designs do more than serve the needs of end users — they embody the needs of every single person (including the designers) involved in their ideation, development, execution and distribution.” should be explained a bit more. It sounds interesting and relevant, but should be understandable to readers who are not deeply involved in practice of design for wellbeing as well.

We understand that the sentence was somewhat ambiguous. We have edited it to make it more clear (we decided not to add additional explanation because we envision that the new version is better embedded in the body of text) (lines 43-45).

2) Differentiation necessary in line 100-101

The following sentence needs a bit more differentiated view: “Moreover, any typology of needs is neither true nor correct as such, because its usefulness depends on the purpose of its application [13].” In a scientific sense a typology should be true. For example, a typology with non-valid needs would be useless. But I agree, that the detailedness and structure of a typology could depend on its usage.

We agree that the sentence in the original manuscript was confusing (our intention was indeed to refer specifically to the granularity). Since we have a much more elaborate discussion about the quality of need typologies a few pages later, we have removed this sentence altogether (lines 100-103).

3) Chapter “4. Developing an Improved Typology of Fundamental Needs”

The process of extending Maslow’s typology based on six further need models is the kernel of the development of the extended framework. The method of doing so is described in just one paragraph (line 329-339). Here some more information on the method and process is necessary:

3a) The guidelines of Brown and Clark and the way they have been applied should be described in some sentences.

We have provided a more precise and elaborate overview of how the guidelines were applied, by specifically referencing their six phases (lines 334-347)

3b) How did the authors cope with differences in interpretation? How big were the differences between the interpretations of the authors? This is relevant, because needs, having the same title in different typologies, are not described in the same way already by the typology authors. How was this solved?

We added two sentences to explain how we dealt with authors differences in perspective (lines 347-351).

3c) Appendix A is very helpful, but it would be more interesting to see the source typology of the needs in table 3. Maybe appendix A could be improved by incorporating table 3 and adding the source typology.

We agree with the suggestion that Appendix A can be improved by incorporating Table 3. We have done accordingly (from line 609 onwards).

4) Grammatic problem in line 513

It seems to be that the sentence in line 513 has a grammatic problem. It is hard to understand.

We have corrected the sentence (line 537).

5) Additional source for line 357 – 360

I agree with the conclusion of the two sentences of line 357 – 360. To make the point stronger it might be helpful to cite Tuch et al. (2016) who tested the Hassenzahl model in leisure and work contexts.

Thank you for the suggestion and for pointing us to the interesting work of Tuch and colleagues. We have added the suggested reference (Tuch et al., 2016) and another reference to a study by Tuch et al. (2013). In addition, we extended the argumentation with a reference to Wiklund-Engblom et al., 2009) (lines 375-378).

Reviewer 2 Report

This is one of those seminal pieces that everyone will cite, either to agree with or to build upon. The one weakness is complexity; the proposed topology is complete, but more complex to apply than what it aims to replace.

At the same time, what an improvement! Many a time I've looked at requirements for a needs assessment, only to have the same questions pop up: is this a need, or just a desire? What's a need, anyway? This work is a terrific step in the direction of consistency. I can see this going over well in my grad and undergrad classes, as well as in professional practice.

Author Response

Thank you very much for your encouraging feedback!

Reviewer 3 Report

This paper displays many strengths. The author(s) present a clearly written, up-to-date interpretation and critique of Maslow’s theory of needs, along with useful descriptions of influential research studies that have led to modifications of the theory. I am also struck by the approach taken to operationalize the updated model using contemporary research that has demonstrated reproducibility. Third, and related to the proceeding, I appreciate the intent of the authors to make their recommendations “teachable” and relevant to design students using vivid examples. I enjoyed reading this paper very much.

I have two suggestions for the author(s) but these should be interpreted as suggestions because they may go beyond what was intended in this paper

  1. Models of well-being and decision-making have gone beyond individual conceptions of well-being, and are considering the role of social partners (pairs or groups) in negotiating shared well-being and understandings. Although I believe that the author(s) touch on this idea in the discussion, I believe it would have been useful to have mentioned this approach in the development of the paper. Work in cross-cultural psychology provides important insights here.
  2. Contemporary models of well-being and self-expression are also more influenced by a life span perspective (continuity and change in preferences, behaviors, activities) and a life course perspective (well-being and “needs” influenced by changes in context over time). The emphasis of Maslow and Maslovian approaches focuses on the commonalities across time and place, while life span and life course perspectives imply that “needs” shift and change, and along with them the maximization of activities to preserve well-being. Laura Carstensen’s work on socio-emotional selectivity is a good example of this approach to well-being and its maximization.

Author Response

We thank the reviewers for their careful reading of our manuscript and their insightful comments and suggestions. Below we respond to the comments of Reviewer 3 in detail (with the original reviewer comments in bold). Note that the line numbers in our responses refer to the lines in the revised manuscript (with track changes), which may differ from the original manuscript.

I have two suggestions for the author(s) but these should be interpreted as suggestions because they may go beyond what was intended in this paper

1) Models of well-being and decision-making have gone beyond individual conceptions of well-being, and are considering the role of social partners (pairs or groups) in negotiating shared well-being and understandings. Although I believe that the author(s) touch on this idea in the discussion, I believe it would have been useful to have mentioned this approach in the development of the paper. Work in cross-cultural psychology provides important insights here.

We agree that it is important to more explicitly stress the social aspects of well-being. We extended the general discussion to nuance our original statement (lines 555-566), and we have extended the analysis of Maslow’s theory, mentioning the need to move beyond  a strictly individual to include also a social version of well-being (lines 236-240 & 269-270). In these discussions we have added additional references to determinants/dimensions of social well-being.

2) Contemporary models of well-being and self-expression are also more influenced by a life span perspective (continuity and change in preferences, behaviors, activities) and a life course perspective (well-being and “needs” influenced by changes in context over time). The emphasis of Maslow and Maslovian approaches focuses on the commonalities across time and place, while life span and life course perspectives imply that “needs” shift and change, and along with them the maximization of activities to preserve well-being. Laura Carstensen’s work on socio-emotional selectivity is a good example of this approach to well-being and its maximization.

Thank you for suggesting the relevance of a life-span perspective. We have added some sentences to highlight the importance of life-span theories of motivation in the general discussion (lines 527-533).